# A rice QTL *GS3.1* regulates grain size through metabolic-flux distribution between flavonoid and lignin metabolons without affecting stress tolerance

Yi-Min Zhang[1,2], Hong-Xiao Yu[1,2], Wang-Wei Ye[1], Jun-Xiang Shan [1], Nai-Qian Dong [1], Tao Guo[1], Yi Kan[1,2], You-Huang Xiang[1,2], Hai Zhang[1,3], Yi-Bing Yang[1,2], Ya-Chao Li[1,3], Huai-Yu Zhao[1,2], Zi-Qi Lu[1,3], Shuang-Qin Guo[1,2], Jie-Jie Lei[1,2], Ben Liao[1,3], Xiao-Rui Mu[1,2], Ying-Jie Cao[1,2], Jia-Jun Yu[1,3] & Hong-Xuan Lin [1,2,3,4✉]

Grain size is a key component trait of grain weight and yield. Numbers of quantitative trait loci (QTLs) have been identified in various bioprocesses, but there is still little known about how metabolism-related QTLs influence grain size and yield. The current study report *GS3.1*, a QTL that regulates rice grain size via metabolic flux allocation between two branches of phenylpropanoid metabolism. *GS3.1* encodes a MATE (multidrug and toxic compounds extrusion) transporter that regulates grain size by directing the transport of *p*-coumaric acid from the *p*-coumaric acid biosynthetic metabolon to the flavonoid biosynthetic metabolon. A natural allele of *GS3.1* was identified from an African rice with enlarged grains, reduced flavonoid content and increased lignin content in the panicles. Notably, the natural allele of *GS3.1* caused no alterations in other tissues and did not affect stress tolerance, revealing an ideal candidate for breeding efforts. This study uncovers insights into the regulation of grain size though metabolic-flux distribution. In this way, it supports a strategy of enhancing crop yield without introducing deleterious side effects on stress tolerance mechanisms.

[1] National Key Laboratory of Plant Molecular Genetics, CAS Centre for Excellence in Molecular Plant Sciences and Collaborative Innovation Center of Genetics & Development, Shanghai Institute of Plant Physiology and Ecology, Chinese Academy of Sciences, Shanghai 200032, China. [2] University of the Chinese Academy of Sciences, Beijing 100049, China. [3] School of Life Science and Technology, ShanghaiTech University, Shanghai 201210, China. [4] Guangdong Laboratory for Lingnan Modern Agriculture, Guangzhou 510642, China. ✉email: hxlin@cemps.ac.cn

Rice, as one of the most important global staple crops, provides sustenance for more than half of the world's population. Rice yield has been determined to be a quantitative trait regulated by a series of QTLs[1–3]. Many yield-related genes were discovered by map-based cloning and mutant identification in the past studies, including genes involved in multiple agricultural traits such as effective tiller number, grain number per panicle, grain weight, and even stress resistance[4–22]. Large numbers of genes were identified and characterized in rice that influence grain weight, which were attached to some important biological pathways[4–6,9,14,16,18,23–28]. GW2 (LOC_Os02g14720) was reportedly involved in the ubiquitin-proteasome pathway regulating grain width[26]. GS3 (Os03g0407400) encodes a protein considered to take part in the G-protein signaling pathway[9,23,27,28]. GSN1 (LOC_Os05g02500) is a mitogen-activated protein kinase phosphatase that plays a role in the mitogen-activated protein kinase signaling pathway[24]. GL3.1 (LOC_Os03g44500) regulates grain length by regulating the cell cycle[5,14,25]. Phytohormones also play a vital role in regulating grain size. For example, GS2, also referred to as OsGRF4 (LOC_Os02g47280), is a transcriptional regulator and participates in the brassinolide pathway[4,16,18].

The phenylpropanoid metabolic pathway, identified to synthesize flavonoids, lignin, and anthocyanins in plants, is important for plant growth and stress tolerance. Increased flavonoids content can enhance stress tolerance such as drought tolerance and UV tolerance in plants[29,30]. NARROW AND ROLLED LEAF 2 (NRL2, LOC_Os03g19520) mutants exhibit a narrow leaf and reduced grain size and lignin content[31]. SQUAMOSA PROMOTER BINDING PROTEIN-LIKE 9 (SPL9, LOC_Os05g33810) regulates stress tolerance by controlling anthocyanin metabolism[32]. Recent study indicated that a UDP-glucosyltransferase, Grain Size and Abiotic stress tolerance 1 (GSA1, LOC_Os03g55040), synergistically regulated grain size and stress tolerance by controlling flavonoid and lignin content[6]. Nevertheless, the knowledge about QTLs that influence how metabolism, especially phenylpropanoid metabolism, influence grain size and yield is still much to be known.

Multidrug and toxic compound extrusion (MATE) transporters were reported to transport a variety of compounds and ions via membrane transport, and to regulate diverse bioprocesses in plants[33]. The best-known process to be regulated by the MATE transporter was stress tolerance, in which large numbers of MATEs were indicated to take part in refs. [34–38]. For example, EDS5 is involved in salicylic acid (SA) biosynthesis, and DTX50 and its rice ortholog DG1 function as an abscisic acid (ABA) transporter[34,35,39]. Some MATEs were also reported to regulate plant development such as lateral organ size and initiation rate[40]. MATE transporter substrates and the mechanism by which they regulate plant development still needs more research. Meanwhile, MATEs encoded within QTLs to influence grain size and yield can be further explored.

The current study cloned and characterized a QTL in rice that regulates flavonoid and lignin metabolic fluxes via the allocation of the metabolic intermediate, p-coumaric acid, which finally influenced grain size and yield in rice.

## Results

### GS3.1 is a QTL for grain size.
One of the chromosome segment substitution lines (CSSLs; the donor parent: HP1, HP for short, Oryza glaberrima; the recurrent parent: HJX74, HJX for short, Oryza sativa indica), HPC078, was identified exhibiting difference in grain size when compared to the recurrent parent, HJX (Supplementary Fig. 1a). A QTL was mapped to the short arm of chromosome 3 and was named GS3.1 for further characterization. GS3.1 showed the effect on grain size, which was determined by 1000-grain weight, grain length, and grain width. The results showed that GS3.1 explained 33.2% of the phenotypic variation for 1000-grain weight, 23.6% of the phenotypic variation for grain length and 23.5% of the phenotypic variation for grain width (Supplementary Table 1).

To clone the individual gene controlling grain size, high-resolution mapping was performed and the GS3.1 locus was narrowed to a 9.45 kbp region (Fig. 1a), and a nearly isogenic line (NIL) of GS3.1, NIL-GS3.1HP was constructed together with a NIL-GS3.1HJX as control (Supplementary Fig. 1b, c). Examining the rice genome database, there was one transcript corresponding to Loc_Os03g12790 that was annotated as a MATE transporter family protein in the GS3.1 region. Sequence comparison between the HJX allele and the HP allele indicated the presence of five nucleotide substitutions and one insertion (G159T, A279G, C921T, C1881T, A1891C, and 1727_1728 ins TGCGGCTGC), which caused one amino acid substitution (N631H) and one insertion (576_577 ins CGC) in Loc_Os03g12790 (Fig. 1a).

To verify that Loc_Os03g12790 was the specific gene that regulated grain size within the GS3.1 locus, Loc_Os03g12790 was knocked out using CRISPR/Cas9 in WYJ7 (9522, Oryza sativa japonica) and the recurrent parent HJX background. The current study observed a significant increase in 1000-grain weight (33.36%), grain length (11.59%), and grain width (14.47%) in HJX background mutants without noticeable changes to plant height and tiller number. Similar phenotypes were observed on grain size in the 9522 background (Fig. 1b–e; Supplementary Fig. 2a–e and Supplementary Table 2). Furthermore, a genetic complementation assay was performed by transforming the Loc_Os03g12790 genic region from HJX with its upstream 2.5 kbp promoter region and 1.5 kbp downstream region into NIL-GS3.1HP. The transgenic construct lines showed a decrease in 1000-grain weight (−9.35%), grain length (−2.03%), and plant height (−5.03%), but there were no significant differences in grain width and tiller number compared to NIL-GS3.1HP controls (Fig. 1f-i and Supplementary Fig. 2f, g). Overexpressing the Loc_Os03g12790 gene region using the rice ubiquitin promoter caused a decrease in grain length together with plant height (Supplementary Fig. 2h–j). These data indicate that Loc_Os03g12790 negatively regulated grain size in the GS3.1 locus of the rice genome.

### GS3.1 enlarges grain size and enhances rice yield by influencing cell expansion.
To further characterize this QTL, the agronomic phenotypes were evaluated. NIL-GS3.1HP exhibited an increase in 1000-grain weight (6.2%), grain length (1.9%), and grain width (2.5%) compared to NIL-GS3.1HJX (Fig. 2a–d, Supplementary Fig. 3a, b, and Supplementary Table 3). Moreover, the current study observed more tillers but no difference in spikelet number or plant height in NIL-GS3.1HP compared to NIL-GS3.1HJX (Fig. 2e, f, Supplementary Fig. 3c, d and Supplementary Table 3). These factors led to an increase in grain yield per plant (16.7%), grain yield per panicle (11.5%) and ultimately plot yield (8.7%) compared to NIL-GS3.1HJX (Fig. 2g–i and Supplementary Table 3).

Next, the current study investigated grain milk filling rate in NIL-GS3.1HP and NIL-GS3.1HJX, and found that there were significant differences between NIL-GS3.1HP and NIL-GS3.1HJX in both wet weight and dry weight of caryopses in the late filling stage. However, there was no significant difference in grain weight from the flowering day to 15 days after flowering (Fig. 2j). These results indicate that the difference in 1000-grain weight might mainly come from an increase in dry matter resulting from a larger spikelet hull.

To assess the contribution of the spikelet hull to grain weight, the cell number and size in young and mature spikelet hulls were investigated by measuring the size of the outer epidermal cells.

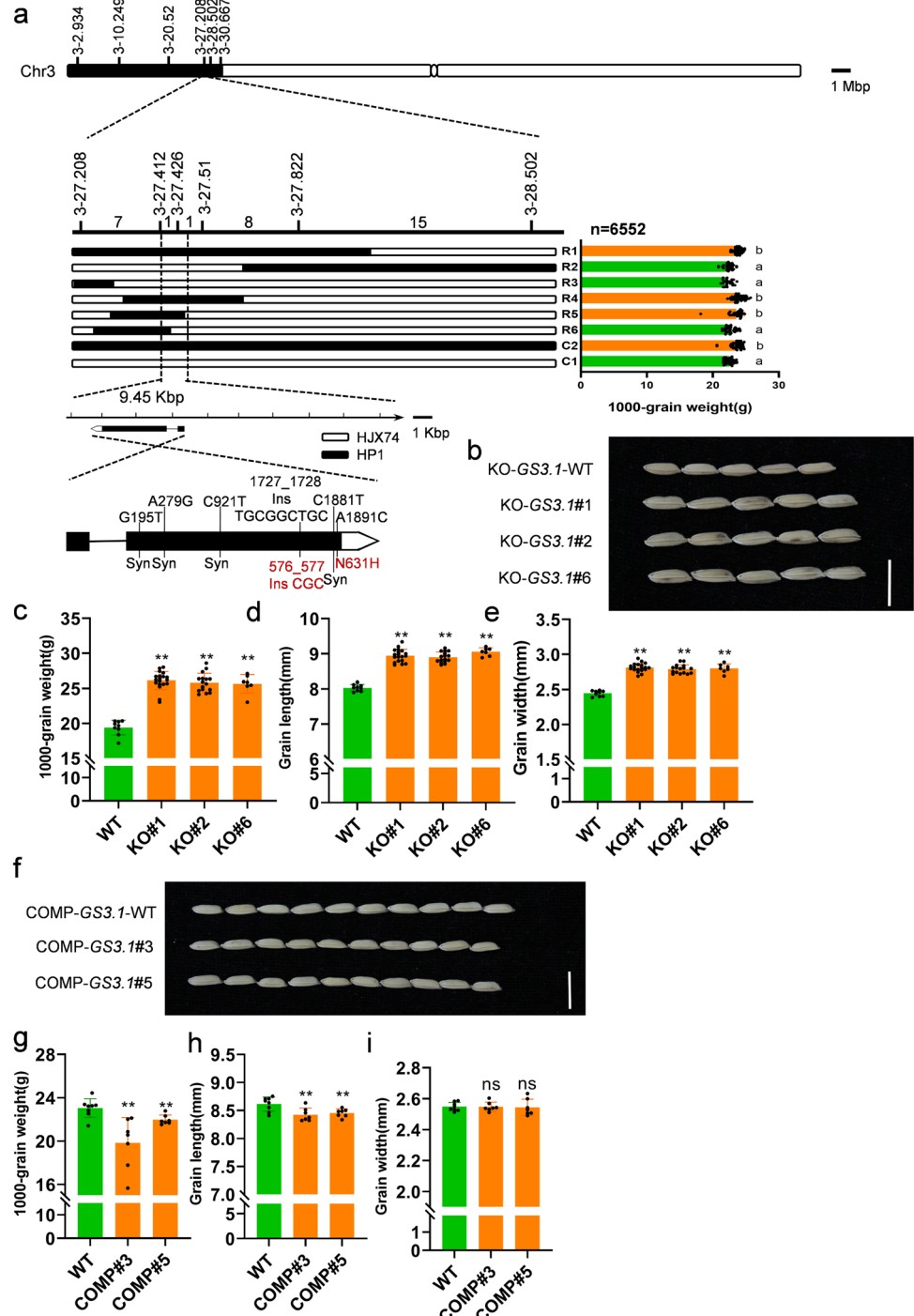

**Fig. 1 Map-based Cloning of *GS3.1*. a** *GS3.1* was preliminarily mapped to the interval between the markers 3-27.208 and 3-28.502 on the short arm of chromosome 3 and then narrowed to a 9.45 kbp region containing one gene. The numbers of recombinant individuals are shown between the marker positions. 1000-grain weight is shown on the right of each representative recombinant line. Recombinant lines R1 to R6 originated from recombinant line C2. Values represent the mean ± s.d. ($n \geq 23$ plants). Different letters indicate significant differences ($P < 0.05$) as determined by a Tukey's test. Single nucleotide mutations and the corresponding amino acid changes in NIL-*GS3.1*[HP] are shown on the schematic illustration of the *GS3.1* gene. Syn, synonymous variations. **b** Mature grains of *GS3.1* knock out lines KO-*GS3.1*#1, KO-*GS3.1*#2, KO-*GS3.1*#6, and the wild-type control. Scale bar =1 cm. Comparison of 1000-grain weight (**c**), grain length (**d**), and grain width (**e**) between *GS3.1* knock out lines and the wild-type control ($n \geq 7$ plants). **f** Mature grains of *GS3.1* complementation lines COMP-*GS3.1* #3, COMP-*GS3.1* #5, and NIL-*GS3.1*[HP] wild-type control. Scale bar =1 cm. Comparison of 1000-grain weight (**g**), grain length (**h**), and grain width (**i**) between *GS3.1* complementation lines and NIL-*GS3.1*[HP] wild-type control ($n \geq 7$ plants). The values in **c–e** and **g–i** represent the mean ± s.d.. *$P < 0.05$ and **$P < 0.01$ indicate significant differences compared to the control in two-tailed Student's *t*-tests. The source data underlying Fig. 1a, c–e and g–i are provided as Source data file.

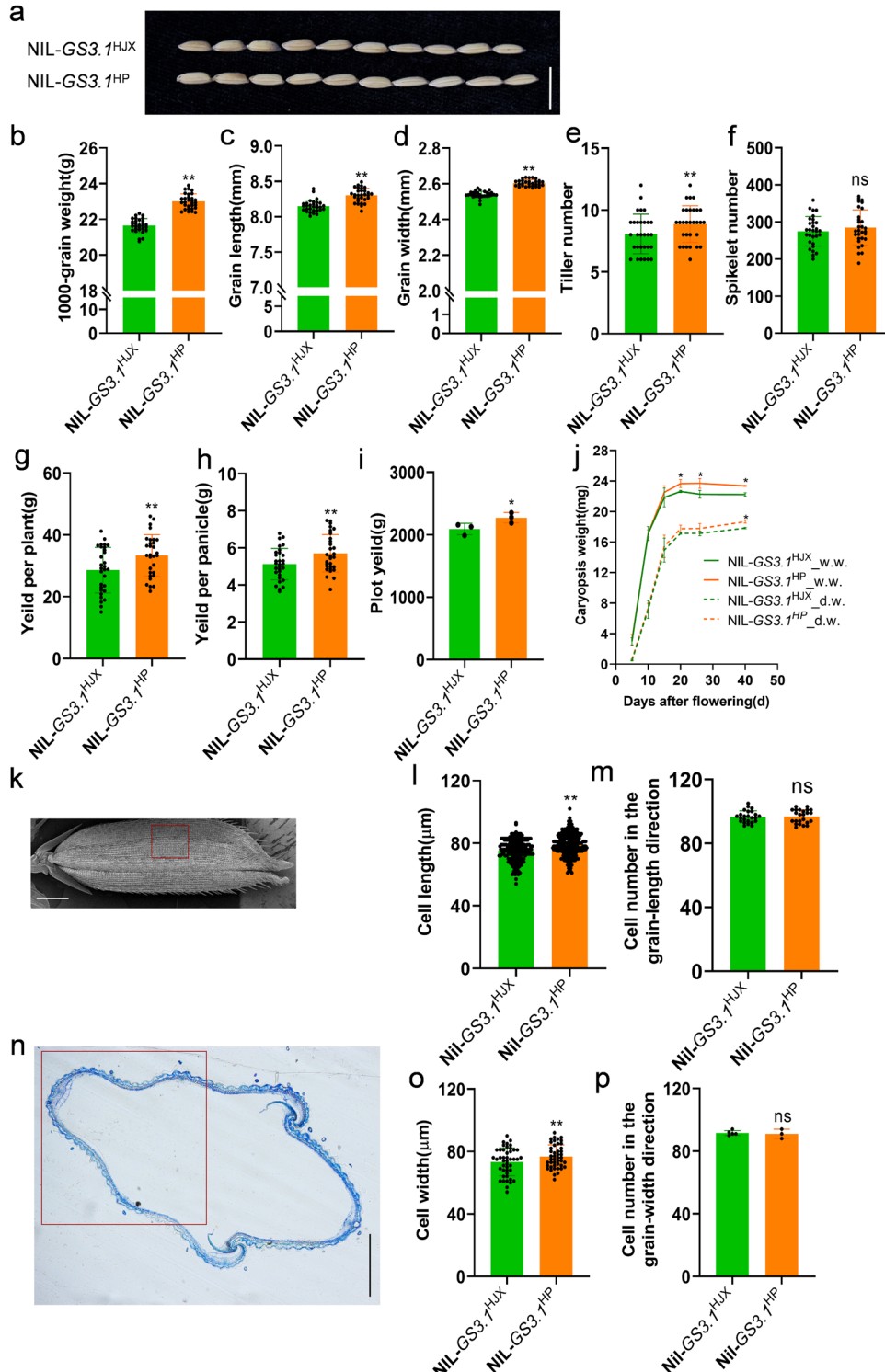

Scanning electron micrographs showed that the cell number was similar between NIL-*GS3.1*[HP] and NIL-*GS3.1*[HJX] in the grain-length direction, but the cell length of spikelet hulls of NIL-*GS3.1*[HP] was significantly increased compared to NIL-*GS3.1*[HJX] (Fig. 2k–m and Supplementary Table 4). Concerning cell width, similar trends were found as the cell number showed no significant difference but the cell width increased in the cross-sections of spikelet hulls (Fig. 2n–p and Supplementary Table 4). Scanning electron micrograph analysis was also performed on the knockout lines of *GS3.1*, KO-*GS3.1*, and the wild-type control. KO-*GS3.1* revealed an increased cell length

and unchanged cell number in the grain-length direction (Supplementary Fig. 4). These results suggested that *GS3.1* enhanced the cell expansion and enlarged grains. Relying on the enhanced grain size, NIL-*GS3.1*[HP] plants had an increase in yield compared to NIL-*GS3.1*[HJX].

**GS3.1 is expressed in the panicles and the phloem of plant leaves and culms.** A *GS3.1* promoter-driven GUS construct was transformed into ZH11 (*Oryza sativa japonica*) and used to

**Fig. 2 GS3.1 is a QTL that controls the grain size by influencing cell expansion. a** Mature grains of NIL-GS3.1$^{HJX}$ and NIL-GS3.1$^{HP}$. Scale bar =1 cm. Comparison of 1000-grain weight (**b**), grain length (**c**), grain width (**d**), tiller number (**e**), spikelet number (**f**), yield per plant (**g**) and yield per panicle (**h**) between NIL-GS3.1$^{HJX}$ and NIL-GS3.1$^{HP}$ ($n = 30$ plants). **i** Comparison of plot yield between NIL-GS3.1$^{HJX}$ and NIL-GS3.1$^{HP}$ ($n = 3$ plots, about 2 square meters per plot). **j** Time-course of the change in caryopsis wet weight and dry weight for NIL-GS3.1$^{HJX}$ and NIL-GS3.1$^{HP}$ ($n = 3$ plants, ≥20 caryopses per plant). **k** Scanning electron micrographs of the outer epidermal cells of spikelet hulls just before heading. Scale bar =1 mm. The red rectangle border shows the area for measuring the cell length. Comparison of the outer epidermal cell length (**l**) and cell number in the grain-length direction (**m**) between NIL-GS3.1$^{HJX}$ and NIL-GS3.1$^{HP}$ ($n = 24$ spikelet hulls, 10 cells per spikelet hull for cell length measuring). **n** Micrographs of the cross-section of spikelet hulls just before heading in the grain width direction. Scale bar = 1 mm. The red rectangle border shows the area used for measuring the cell width. Comparison of the outer epidermal cell width (**o**) and cell number in the grain-width direction (**p**) between NIL-GS3.1$^{HJX}$ and NIL-GS3.1$^{HP}$ ($n ≥ 3$ spikelet hulls, $n = 45$ cells). The values in **b–h**, **j**, **l**, **m**, **o**, **p** represent the mean ± s.d. *$P < 0.05$ and **$P < 0.01$ indicate significant differences compared with NIL-GS3.1$^{HJX}$ in two-tailed Student's $t$-tests. The values in **i** represent the mean ± s.d.. *$P < 0.1$ indicate significant differences compared to NIL-GS3.1$^{HJX}$ in two-tailed Student's $t$ tests. The source data underlying Fig. 2b–j, l, m, o and p are provided as Source data file.

analyze the expression pattern of *GS3.1*. The panicles, spikelet hulls, leaves, and culms were stained blue after the incubation with X-gluc (Fig. 3a–c, and Supplementary Fig. 5a–c). Upon further observation, the current study found that the blue stain in culms, leaves and late-stage panicles had a consistent localization within the vasculature. So that, cross-sections of stained young leaves were observed and the phloem was found to be stained (Fig. 3d, and Supplementary Fig. 5d). Then, the current study examined the tissue-specific expression pattern of *GS3.1* using qRT-RCR and revealed a similar pattern as the GUS staining assay. *GS3.1* was wildly expressed in various organs including panicles, leaves, culms and culm nodes, but barely expressed in roots. The relative expression levels of *GS3.1* in NIL-GS3.1$^{HJX}$ were significantly higher compared to that of NIL-GS3.1$^{HP}$ in young panicles (3–5 cm) and were slightly higher in culms but showed no significant difference in other tissues (Fig. 3e).

Previous studies showed that the ortholog of *GS3.1* in *Arabidopsis* (*ABS4*) had five paralogs (*ABS3*, *ABS3L1*, *ABS3L2*, *ABS3L3*, and *ABS3L4*) that might have similar function[41]. A maximum-likelihood phylogenetic tree was constructed with the amino acid sequences of all six *Arabidopsis* genes (*ABS3*, *ABS3L1*, *ABS3L2*, *ABS3L3*, *ABS3L4*, and *ABS4*), their corresponding orthologs in rice (*Loc_Os06g36530*, *Loc_Os04g48290*, and *GS3.1*), and an *Arabidopsis* MATE gene (*DTX14*) as the outgroup (Supplementary Fig. 6a). All nine genes (excluding the outgroup) were divided into four classes, and three rice paralogs separated into three classes, indicating that these three rice paralogs including *GS3.1* may have functional redundancy in rice. Relative expression levels of these genes were determined using qRT-PCR to examine the expression pattern, and *Loc_Os04g48290* was the most highly expressed among the three *MATE* paralogs. The expression level of *Loc_Os04g48290* was over 10 times higher compared to *GS3.1* in other tissues except in panicles, indicating that *Loc_Os04g48290* had a more important role in these tissues (Supplementary Fig. 6b). When it comes to panicles, *Loc_Os04g48290* and *GS3.1* showed a comparable expression level which indicated that *GS3.1* contributed a relatively more important role in panicles than in other tissues (Supplementary Fig. 6b). The expression level of *Loc_Os06g36530* was not detected in any of the analyzed tissues (Supplementary Fig. 6b). With the evidence above, the current study concluded that *GS3.1* was expressed in various tissues including panicles, leaves, and culms, but only at a minimal level in roots. In mature tissues, *GS3.1* was expressed specifically in the phloem. NIL-GS3.1$^{HP}$ showed a decreased expression level of *GS3.1* only in young panicles compared to NIL-GS3.1$^{HJX}$, indicating that *GS3.1* had a more prominent function in young panicles to regulate grain size. Two paralogs of *GS3.1* might functionally complement *GS3.1*, but in panicles, *GS3.1* likely to serve a more prominent role because of the lower relative expression level of its paralogs compared to their relative expression in the other tissues.

**GS3.1 functions as a MATE transporter and might transport *p*-coumaric acid in rice.** As *GS3.1* is annotated as a MATE transporter, an *E. coli* growth complementation assay was used to affirm that *GS3.1* functioned as a multidrug efflux to complement the function of *Acrb*[42]. The current study expressed the *GS3.1* full-length CDS by a *pMAL-c5x* vector (pMAL) in an *acrb* loss-of-function *E. coli* mutant, BW25113Δ*acrb*, and observed its growth on the medium supplemented with the antibiotic norfloxacin, on which the control stain transformed pMAL could not grow, confirming that GS3.1 has similar function with its ortholog MATE transporter acrB in *E. coli*. (Fig. 4a and Supplementary Fig. 7).

Comparing to some MATE transporters with solved crystal structures, such as DTX14 in *Arabidopsis*, the structure of GS3.1 was predicted. Similar to those known MATE proteins, GS3.1 had twelve transmembrane helixes (TM1-12) that formed an internal cavity. Twelve helixes arranged into two bundles (TM1-6 and TM7-12), forming a typical V-shape conformation. TM7 is in the middle of this conformation and was supposed to have a major effect on the predicted transporting function (Fig. 4b, c)[43]. As the GS3.1 transmembrane domain above-mentioned is similar to those structurally available MATEs, the transmembrane domain (amino acids 45-496) of the predicted structure was chosen for molecular docking studies with candidate compounds that might be transported by GS3.1. Finally, *p*-coumaric acid was considered to be a top candidate compound. The binding free energy of *p*-coumaric acid was calculated to be −91.36 kcal·mol$^{-1}$ docking with the transmembrane domain of GS3.1. Six amino acid residues in TM7, TM10, and TM11 were important for establishing the interaction of *p*-coumaric acid for binding free energy calculated under −4 kcal·mol$^{-1}$ (Fig. 4c). To conclude, GS3.1 works as a MATE transporter, and it is predicted to transport *p*-coumaric acid as one of its substrates.

**GS3.1 regulates grain size by modulating flavonoid and lignin biosynthesis.** To further illuminate the effect of *GS3.1*, widely-targeted metabolomic analysis was applied on KO-GS3.1 (9522) and the wild-type control. *P*-coumaric acid was proved to be the common substrate of flavonoid and lignin biosynthesis. *P*-coumaric acid can be transformed to *p*-coumaroyl-CoA by *4CLs*, and then to naringenin chalcone by the catalysis of *CHS*. Naringenin chalcone can then be transformed into naringenin and other flavonoids[44,45]. On the other side, *p*-coumaric acid can also be transformed into a set of phenolic acids such as caffeic acid and ferulic acid. Then, these can be transformed into coniferyl alcohol and lignans, which is a substrate for lignin biosynthesis[46]. The widely-targeted metabolome results showed that in the panicles of KO-GS3.1, *p*-coumaric acid content was reduced along with one main flavonoid, naringenin, but levels for caffeic acid and ferulic acid were elevated along with their downstream product coniferyl alcohol and one kind of lignans,

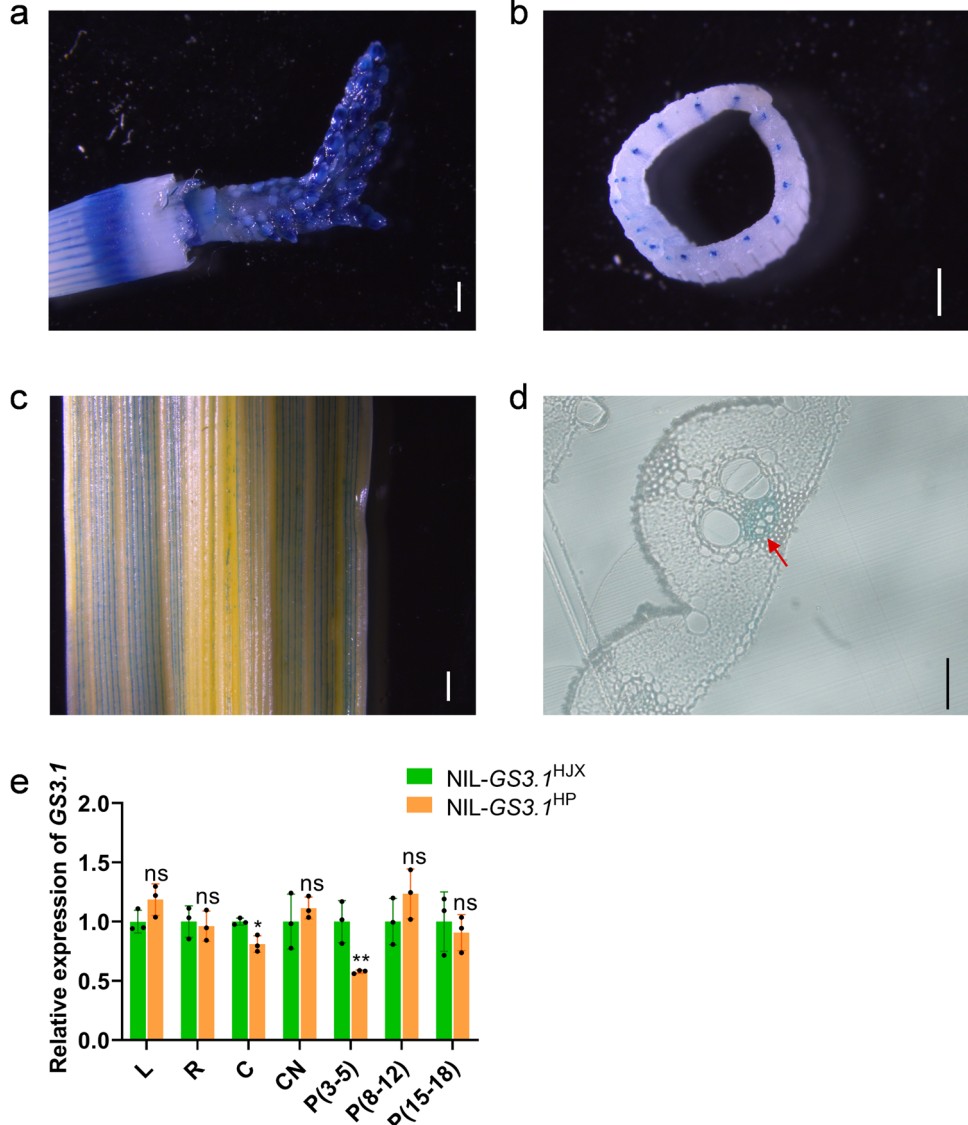

**Fig. 3 GS3.1 expresses in young panicles and the phloem of leaves and culms.** GUS staining of *Pro GS3.1*^HJX:*GUS* of a young 1 cm-long panicle (**a**), culm (**b**), and leaf (**c**). Scale bar = 1 mm. **d** Micrograph of a cross-section of a young leaf. The red arrow shows the stained phloem. Scale bar = 50 μm. **e** The relative expression levels of *GS3.1* in NIL-*GS3.1*^HJX and NIL-*GS3.1*^HP leaves (L), roots (R), culm nodes (CN), culms (C), and panicles (P, numbers indicate the length of young panicles, cm) as determined by qRT-PCR (*n* = 3 biological replicates). The actin gene was used for normalization. The values in **e** represent the mean ± s.d.. *$P < 0.05$ and **$P < 0.01$ indicate significant differences compared to NIL-*GS3.1*^HJX in two-tailed Student's *t*-tests. The source data underlying Fig. 3e are provided as Source data file.

secoisolariciresinol-4-glucoside (Fig. 5a–f). Further tests illustrated that in KO-*GS3.1* the flavonoid content was downregulated and lignan content was upregulated on the whole (Fig. 5g, h). This tendency was also consistent in the comparison of NIL-*GS3.1*^HP and NIL-*GS3.1*^HJX (Supplementary Fig. 8). These data indicated that *GS3.1* might be involved in flavonoid biosynthesis by transporting *p*-coumaric acid. Consequently, downregulation of *GS3.1* caused the suppression of flavonoid biosynthesis, meanwhile, lignin biosynthesis was accelerated.

To further verify the effects on lignin production caused by *GS3.1*, the lignin content was examined using a *GS3.1* overexpression line, OE-*GS3.1*, and the lignin content of OE-*GS3.1* was observed to be decreased compared to its wild-type control (Fig. 5i). The relative expression levels of genes participating in the biosynthesis pathway mentioned above were examined in the panicles. The current study found that in

KO-*GS3.1*, flavonoid-biosynthesis-related genes (*CHS*, *F3H*, and *F3'H*) were downregulated, as well as genes participating in both flavonoid and lignin biosynthesis (*4CL1*, *4CL3*, *4CL4*, and *4CL5*) compared to the wild-type control (Fig. 5j). Previous studies indicated that flavonoids could suppress auxin transport, biosynthesis, and the subsequent signal transduction[47], this phenomenon could also be observed in *KO-GS3.1* which had a lower flavonoid level and showed higher expression of the auxin transporter *PIN1b*, auxin synthase *YUCCA7*, auxin response factors (ARFs; *ARF15* and *ARF24*) and a lower expression level of the auxin inactivation protein *GH3.8*. GA signaling is considered to be activated by auxin by promoting its synthesis and suppressing its inactivation[48], so one GA inactivation enzyme *GA2ox10*, and two grain size-related GA downstream genes, *GRF1* and *GRF6* were also be examined. The results showed that *GA2ox10* was downregulated and *GRF*s were

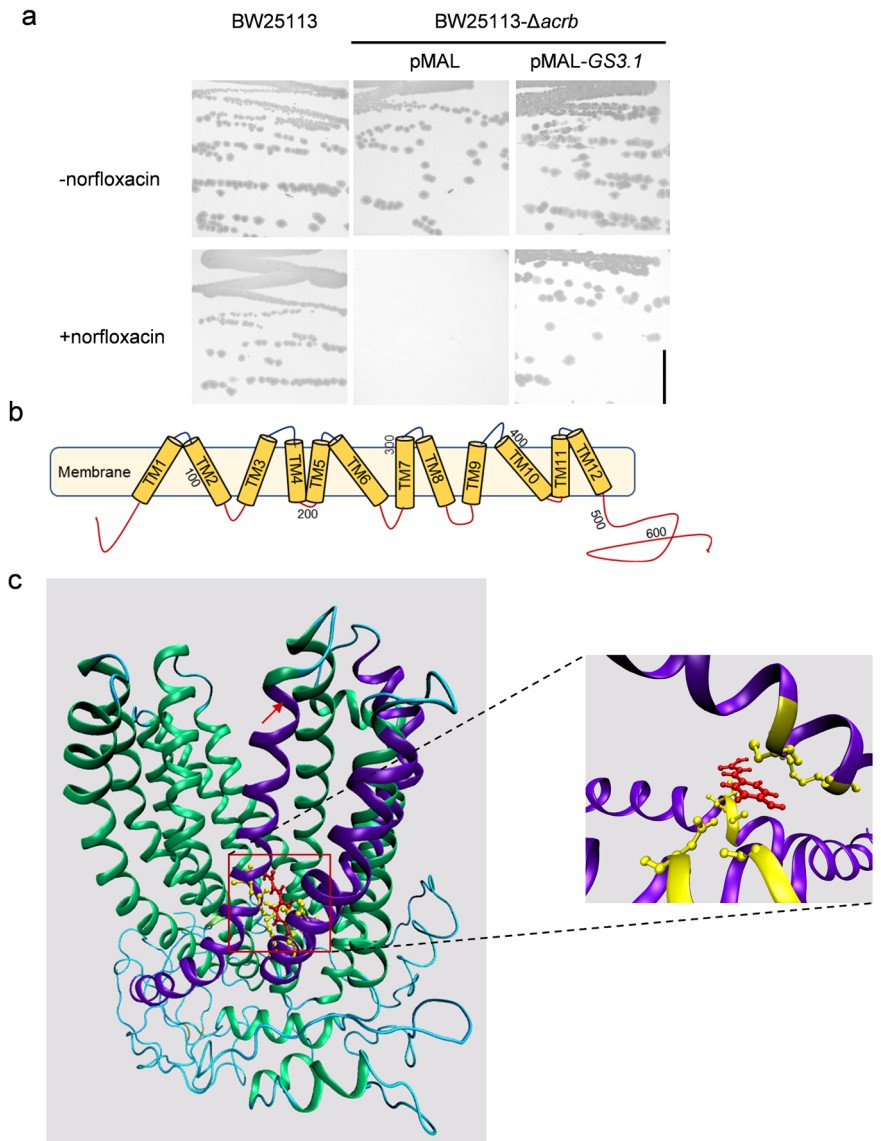

**Fig. 4 *GS3.1* functions as a MATE transporter and might transport *p*-coumaric acid in rice. a** Growth complementation assays of the *E. coli* Δ*acrB* mutant expressing *GS3.1*[HJX] on medium supplemented with the antibiotic norfloxacin. The *E. coli* Δ*acrB* mutant expressing pMAL empty vector and the wild-type *E. coli* BW25113 were used as control strains, and mediums without norfloxacin were used as the growth condition control. Scale bar = 1 cm. **b** A schematic of the predicted *GS3.1* structure showing 12 transmembrane helixes. Numbers indicate the sequence number of amino acid residues. **c** Predicted *GS3.1* model docked with *p*-coumaric acid (red). The α-helices interacting with *p*-coumaric acid were colored purple. The amino residues interacting with *p*-coumaric were colored yellow. The red arrow indicates the key α-helix for its transporter function.

upregulated in KO-*GS3.1* compared to the wild-type control (Supplementary Fig. 9a). The same assay was examined in NIL-*GS3.1*[HP] and NIL-*GS3.1*[HJX], and the tendency of the above-mentioned marker genes was consistent with the comparison of KO-*GS3.1* and the wild-type control (Supplementary Fig. 9b, c). All of the above results indicated that *GS3.1* might participate in flavonoid and lignin biosynthesis by regulating the allocation of *p*-coumaric acid. The altered flavonoid and lignin biosynthesis influence downstream auxin and GA signaling pathways and cell wall synthesis, thereby resulting in a change of cell size and grain size.

## Discussion

Enlarged grain size can directly enhance rice yield. As such, grain size became one of the most-studied yield traits over the past few decades. There were many QTLs discovered that regulated grain size though various pathways[4–6,9,14,16,18,23–28]. For example, QTLs involved in grain size has been linked to the ubiquitin-proteasome pathway, G-protein signaling pathway, mitogen-activated protein kinase signaling pathway, phytohormone pathways, and transcriptional regulation[2]. In recent years, some metabolism-related genes were linked to grain size such as *Os02g57760* and *GSA1*. The former regulated grain size by affecting the trigonelline pathway and the latter by the phenyl-propanoid pathway[6,49]. These two genes functioned as enzymes that directly catalyze their substrates, but another way to influence metabolism is to change the distribution of metabolic intermediates. For example, EDS5 transports isochorismate from plastids to the cytosol for further biosynthesis of SA[34]. The present study identified a QTL, *GS3.1*, which functioned as a MATE transporter that may transport a critical metabolic intermediate,

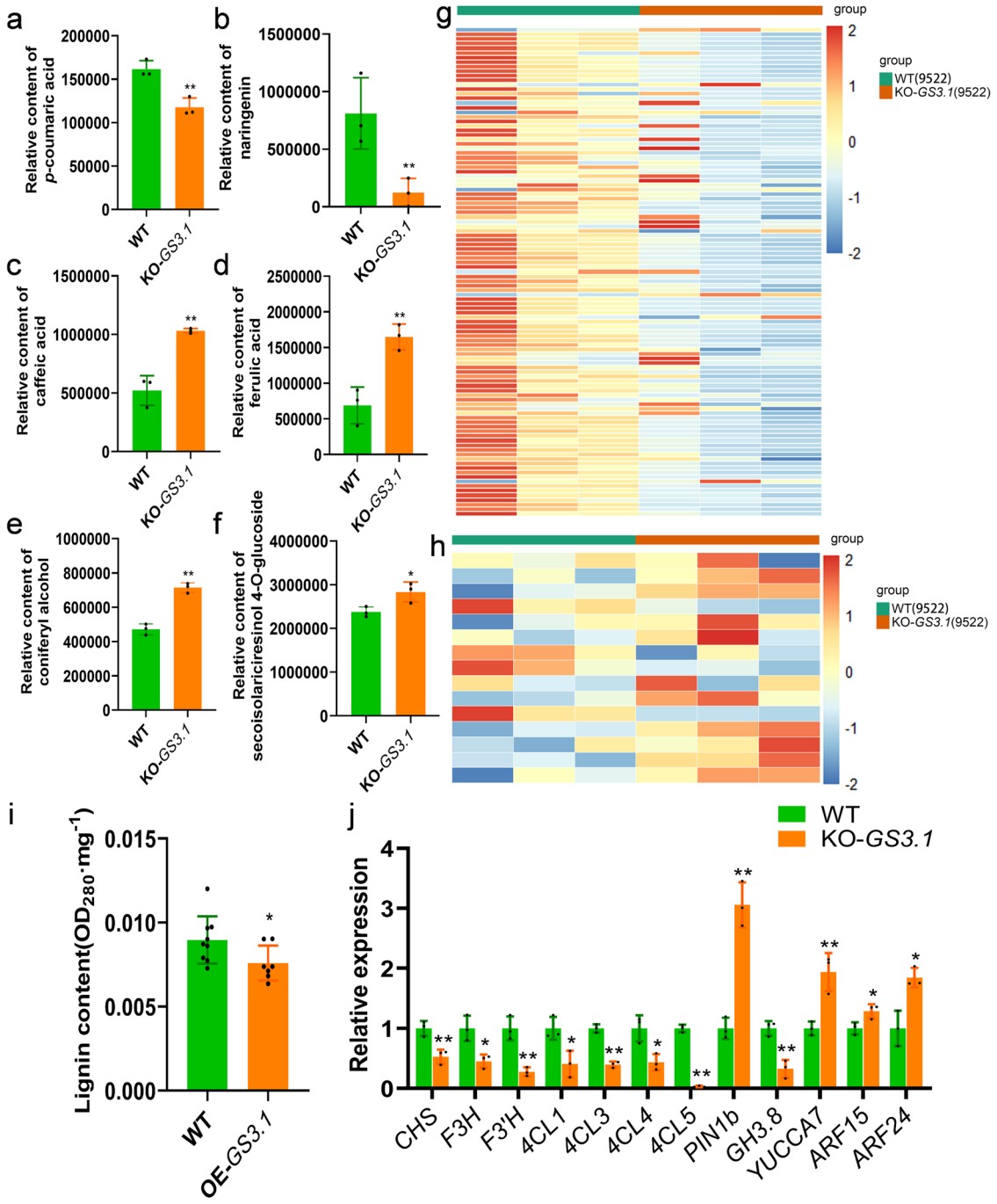

**Fig. 5 GS3.1 may participate in the flavonoid biosynthesis pathway leading to changes in lignin biosynthesis and the flavonoid-mediated auxin pathway.** Comparison of the relative content of *p*-coumaric acid (**a**), naringenin (**b**), caffeic acid (**c**), ferulic acid (**d**), coniferyl alcohol (**e**), and secoisolariciresinol 4-O-glucoside (**f**) between the *GS3.1* knock-out line KO-*GS3.1* panicles and wild-type control panicles (n = 3 biological replicates, 4 plants per biological replicate). Heat maps of the relative content of flavonoids (**g**) and lignans (**h**) in KO-*GS3.1* panicles and wild-type control panicles determined using metabolomics. Standard-scores (Z-scores) were used as the numerical metrics to evaluate the standard deviations from the mean of the corresponding samples. **i** Comparison of lignin content between the leaves of *GS3.1* overexpression lines OE-*GS3.1* and wild-type controls (n = 9 leaves from 3 plants for wildtype, n = 7 leaves from 3 plants for OE-*GS3.1*). **j** The relative expression of flavonoid and lignin biosynthesis-related genes and auxin-related genes in *GS3.1* knock-out line KO-*GS3.1* panicles and wild-type control panicles determined by qRT-PCR (n = 3 biological replicates). The values in **a**–**f**, **i** and **j** represent the mean ± s.d. *$P < 0.05$ and **$P < 0.01$ indicate significant differences compared to the wild-type control in two-tailed Student's *t*-tests. The source data underlying Fig. 5a–j are provided as Source data file.

*p*-coumaric acid, and control the metabolic fluxes of flavonoid and lignin biosynthesis by regulating the allocation of their common substrate, *p*-coumaric acid.

Previous studies indicated that auxin played a vital role in grain formation[8,12]. Auxin regulation includes auxin biosynthesis, transport, and signal transduction, which is considered to affect cell expansion[8,12]. *Big Grain1* (*BG1*) is a positive regulator of primary auxin response and auxin transport[12]. Overexpression of *BG1* resulted in larger grains by altering indole-3-acetic acid (IAA) distribution and transport. The mutant of *TILLERING*

*AND SMALL GRAIN 1* (*TSG1*) exhibited small grains by controlling local auxin biosynthesis[8]. *ARFs* such as *OsARF24* were reported to regulate cell elongation in rice[50]. When it comes to *GS3.1*, some auxin-related genes such as *OsYUCCA7*, *OsPIN1b*, and *OsARF24* were upregulated in KO-*GS3.1*, indicating that the auxin pathway was active in the absence of *GS3.1* (Fig. 5j).

For studying the relationship between flavonoid synthesis and auxin transport, previous studies used a series of mutants named *transparent testa* (*tt*) in *Arabidopsis*[47,51–53]. Some of the *tt* mutants (*tt3*, *tt4*, *tt5*, *tt6*, and *tt7*) were known to be related to flavonoid biosynthesis, and *tt4* was the most wildly-studied because it was a loss-of-function mutant of a key enzyme in flavonoid biosynthesis, chalcone synthase[51,52,54]. The *tt4* mutant lacked flavonoids and showed a high rate of auxin transport, which was restored to a normal rate by treating with a flavonoid (naringenin), or flavonoid analog (1-N-naphthylphthalamic acid, NPA)[52,55,56]. The current study showed that in KO-*GS3.1*, the naringenin and total flavonoid content were downregulated compared to the wild-type (Fig. 5b, g). This study supposed that the activation of the auxin pathway is due to the reduced flavonoid content caused by the reduction of GS3.1 function in KO-*GS3.1* and NIL-*GS3.1*[HP]. The decrease in flavonoid content maybe weakened the suppression of auxin transportation in panicles, which further activated the auxin signaling pathway, and finally led to enlarged cell and grain size.

Lignin is considered to be related to plant growth and stress resistance and has been shown to affect plant height, leaf size, and grain size[31,57–60]. Expression of a lignin biosynthetic gene, *CCoAOMT* from jute (*Corchorus capsularis*) in *Arabidopsis* resulted in increased plant height and silique length relative to non-transgenic plants[59]. *NRL2* regulates lignin content and the loss-of-function mutant had narrow leaves and small grains compared to the wild-type[31]. *GSA1* is a QTL that was associated with reduced grain size and lower lignin content[6]. In KO-*GS3.1* the content of the lignin precursor was increased, and the grain size became larger than its respective control (Fig. 5e, f, h and Fig. 1b–e). On the other hand, lignin content was decreased in OE-*GS3.1* which had a reduced grain size compared to its control (Fig. 5i and Supplementary Fig. 2j).

MATE was first characterized as a xenobiotic efflux pump, which was related to multidrug resistance. A series of MATE structures were resolved, within which was a crystal structure of AtDTX14 (c5y50A). AtDTX14 was determined to be the most similar to GS3.1[43,61,62]. Mechanistic studies of AtDTX14 showed that AtDTX14 bonded its substrate at TM7, TM8, and TM10, and that TM7 played a vital role in its transport function[43]. Previous studies have described a large number of MATEs in plants, and clarified diverse secondary metabolites as substrates, including flavonoids, phytohormones, and their precursors, ions, nicotine, proanthocyanidin precursors, and toxic compounds[34,35,37,38,63–68]. Interestingly, in this study, *p*-coumaric acid was predicted to bond GS3.1 at TM7, TM10, and TM11 similarly to the AtDTX14 binding site. Thus, the current study predicted that *GS3.1* likely functions as a MATE transporter for *p*-coumaric acid in rice (Fig. 4c).

Previous studies revealed that part of the phenylpropanoid biosynthetic pathway, including the enzymes producing *p*-coumaric acid from phenylalanine, transforming *p*-coumaric acid to naringenin (a flavonoid) and transforming *p*-coumaric acid to coniferyl alcohol (a monolignol) were localized to the cytoplasm[69]. Further studies indicated that some enzymes in these processes such as hydroxylcinnamoyl transferase (HCT) and cinnamate-4-hydroxylase (C4H) were anchored on the rough endoplasmic reticulum (rER) and some related cytoplasmic enzymes interacted with them to form complexes. For example, phenylalanine ammonia-lyase (PAL), the key enzyme for *p*-coumaric acid synthesis, interacted with C4H during *p*-

coumaric acid biosynthesis process[70–75]. Additional evidence suggested that the rate-limiting enzyme for lignan biosynthesis, coumarate-3-hydroxylase (C3H) interacted with C4H[76]. Early flavonoid and iso-flavonoid biosynthesis enzymes were also reportedly located at the rER[70]. There is a theory that the cooperating enzymes in the phenylpropanoid pathway organized into complexes called metabolons in order to promote activation efficiency[73,74,77]. Based on the aforementioned evidence, the current study considers that the processes of phenylalanine catalysis into *p*-coumaric acid, *p*-coumaric acid catalysis into naringenin, and *p*-coumaric acid catalysis into coniferyl alcohol were separated into three metabolons, and the forepart of lignans biosynthesis metabolon might be more closely associated with the *p*-coumaric acid biosynthesis metabolon compared to flavonoid biosynthesis metabolon because of the direct interaction of C3H and C4H.

Previous studies had indicated that membrane-less compartments were formed based on liquid-liquid phase separation (LLPS), which relies on protein-protein interaction or some other noncovalent interactions to form biomolecular condensates[78,79]. An Autophagy-related protein, ATG8 (AUTOPHAGY-RELATED PROTEIN 8), recognizes and takes part in the protein condensate formation by LLPS[80]. The above-mentioned *GS3.1* ortholog in *Arabidopsis*, *ABS4*, reportedly interacts with *ATG8*[41]. This result was also verified using *GS3.1* and *OsATG8b* by yeast two-hybrid and luciferase complementation assays (Supplementary Fig. 10). There is evidence that *ABS4* localizes to late endosomes[81], and these endosomes were observed to interact with protein condensates formed by LLPS[82]. Taken together, the *p*-coumaric acid biosynthetic metabolon, the lignans biosynthetic metabolon, and the flavonoid biosynthetic metabolon might be separated as three protein condensates formed by LLPS. The lignan biosynthetic metabolon might be closely functionally associated with the upstream *p*-coumaric acid biosynthetic metabolon. However, for flavonoid biosynthetic metabolon, *p*-coumaric acid might be transported to the flavonoid biosynthetic metabolon by GS3.1 with the assist of proteins like ATG8.

*GS3.1* expression was significantly different between NIL-*GS3.1*[HJX] and NIL-*GS3.1*[HP] in panicles compared to other tissues, which indicated that *GS3.1* might be regulated by a panicle-specific transcriptional factor. Meanwhile, the *cis*-element of this transcriptional factor differed between the promoters of NIL-*GS3.1*[HJX] and NIL-*GS3.1*[HP] alleles. Furthermore, the redundant paralogs of *GS3.1* exhibited a different expression pattern compared to that of *GS3.1*, indicating that *GS3.1* played a relatively more important role in panicles than other tissues such as leaves (Supplementary Fig. 6b). The phenylpropanoid pathway affects both plant growth and stress tolerance[6,29–32]. Reduced flavonoid content causes enhanced plant growth but also causes the reduction of stress tolerance as a side effect. The current study found that NIL-*GS3.1*[HP] has decreased *GS3.1* expression only in young panicles but shows no difference in other tissues. Furthermore, there was no observed changes in flavonoid biosynthesis in other tissues and consequently not to cause the stress tolerance reduction (Supplementary Figs. 11 and 12).

To summarize the evidence above, a conceptual model was proposed for the regulation of grain size by modulating the allocation of *p*-coumaric acid between flavonoid and lignin anabolism. Compared to NIL-*GS3.1*[HJX], the expression level of *GS3.1* in NIL-*GS3.1*[HP] is decreased, which decreased the rate of *p*-coumaric acid transport in the panicle. Thus, *p*-coumaric acid is more readily available for the lignan biosynthetic metabolon, which leads to increased lignin biosynthesis and suppression of flavonoid biosynthesis in the panicle. Then, the reduced flavonoid levels weaken the inhibition of auxin transport, biosynthesis, and downstream signal transduction. The activated auxin pathway,

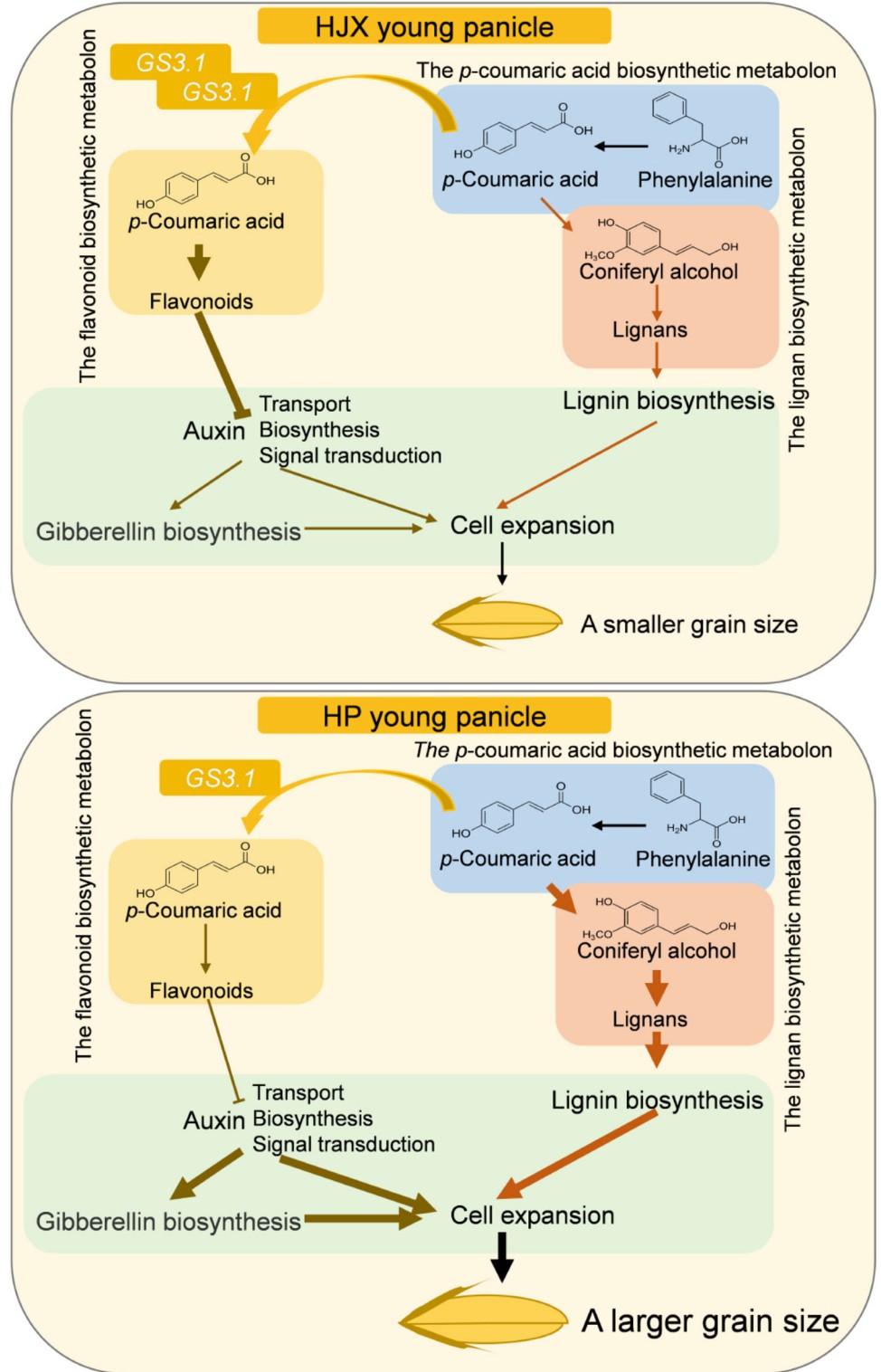

**Fig. 6 Conceptual model of the role of *GS3.1* in the regulation of grain size in rice.** In HP young panicles, a lower *GS3.1* expression level leads to reduced *GS3.1* protein. As a result, less *p*-coumaric acid is transported to the flavonoid biosynthetic metabolon, which reduces flavonoid biosynthesis and instead increases *p*-coumaric acid availability for lignan biosynthesis. The decreased flavonoid content weakens the suppression of the auxin transport, biosynthesis and signal transduction, which in combination with subsequent activation of the gibberellin signaling and the increased lignin content results in a larger grain size in rice.

subsequent activation of the GA pathway, and the increased lignin content encourage cell expansion, thereby enlarging grains and enhancing the rice yield (Fig. 6). Moreover, in NIL-*GS3.1*HP, there was no change in flavonoid biosynthesis in other tissues

such as leaves and consequently not to result the stress tolerance reduction (Supplementary Figs. 11 and 12).

To conclude, the current study reported the identification and characterization of a QTL, *GRAIN SIZE 3.1* (*GS3.1*), which

encodes a MATE transporter protein. GS3.1 was found to mediate the distribution of *p*-coumaric acid between flavonoid biosynthesis and lignin biosynthesis, which led to the differences in flavonoid and lignin content between the panicles of natural allelic variants. Furthermore, flavonoid biosynthesis influenced the downstream auxin pathway and the gibberellic acid (GA) pathway. These alterations together with lignin content influenced grain size. Meanwhile, this study also discovered a gene resource by which rice yield can be enhanced without reducing stress tolerance.

## Methods

**Plant materials and growth conditions.** A set of CSSLs (Chromosome Segment Substitution Lines) was constructed for identifying and characterizing grain size QTLs. An African rice variety, HP1 (HP, *O. glaberrima*, IRGC100127, International Rice Research Institute), was used as the donor parent and an Asian rice variety, HUA-JING-XIAN-74 (HJX74, HJX, *O. sativa indica*, CNA20050527.0, South China Agricultural University), was used as the recurrent parent. One CSSL named HPC078 was selected and backcrossed to HJX for mapping the QTL of grain size, and the QTL was named as GS3.1. The current study then consecutively selected plants that contained the GS3.1 locus and almost all other regions were homozygous for HJX for several rice harvests by marker-assisted selection. Finally, we selected a line that had the smallest region homozygous for HP that also contained GS3.1, referred to as NIL-GS3.1<sup>HP</sup>, and its isogenic control, referred to as NIL-GS3.1<sup>HJX</sup>. All rice plants were cultivated in an experimental field in Shanghai (121°E, 30°N, alt. 4.5 m) during the summer (May–October, 2014–2020, max temperature: 38 °C, min temperature: 17 °C, rainfall: 723.1 mm) and in Hainan (109°E, 18°N, alt. 7 m) during the spring (November– April of the following year, 2015–2020, max temperature: 33 °C, min temperature: 18 °C, rainfall: 152.3 mm) under natural growth conditions. The soil type was waterloggogenic paddy soil, and the soil pH was 6.65 in Shanghai and 5.94 in Hainan. The rice was transplanted 25 days after seeded and harvested at ripening stage (about 110 days after transplanting). Seedlings were fertilized thrice and applied pesticide five times during the grown period.

**Fine mapping of the QTL of grain size.** Grain length, grain width, and 1000-grain weight were measured to characterize grain size during fine mapping of GS3.1. Six molecular markers (3-2.934, 3-10.249, 3-20.52, 3-27.208, 3-28.502, 3-30.667 on chromosome 3) and 200 $F_2$ plants of HPC078 backcrossed to HJX were used for fine mapping and GS3.1 was mapped to the interval between 3-27.208 and 3-28.502. Then, four new molecular markers (3-27.412, 3-27.426, 3-37.51, 27.822 on chromosome 3) in the aforementioned interval were designed and used for detecting recombinants in 6552 $F_2$ plants. GS3.1 was mapped in the interval between the molecular markers 3-27.412 and 3-27.510 and then narrowed to a 9.45 kbp region on the short arm of chromosome 3 by sequencing. The candidate gene, Loc_Os03g12790, was amplified from HPC078 and HJX genomic DNA, sequenced and compared. The PCR primers used in this study are listed in Supplementary Data 2. Linkage maps were constructed using MAPMAKER/EXP 3.0 software[83]. QTLs were identified using the MAPMAKER/QTL program[84].

**RNA extraction and qRT-PCR.** Total RNA was extracted using the E.Z.N.A Plant RNA Kit(Omega). Reverse transcription and cDNA synthesis were performed using the Evo M-MLV RT Kit with gDNA Clean for qPCR (Accurate Biotechnology) from 500 ng total RNA. The ABI 7300 Real-Time PCR System, Fast Start Universal SYBR Green Master MIX with ROX (Roche), and SYBR Green Premix Pro Taq HS qPCR Kit (Accurate Biotechnology) were used to perform quantitative RT-PCR analysis. An Applied Biosystems 7300 Real-Time PCR System Software (Version 1.4.0) was used for data collection and the $2^{-\Delta\Delta CT}$ method and Microsoft Excel 2019 were used for data analysis. Actin (LOC_Os03g50885) was used for normalization and either three or four biological replicates were used for one analysis. PCR primers for qRT-PCR are listed in Supplementary Data 2.

**Plasmid construction.** The CRISPR/Cas9 gene-editing construct of GS3.1 was designed and constructed as per the previous study[85]. Two designed gRNAs of GS3.1 were ligated to the OsU3 and OsU6a promoters and transferred into the pYLCRISPR/Cas9-MTmono vector. The plasmids for split-ubiquitin assay were constructed using T4 DNA ligase (New England Biolabs). The CDS fragments were prepared by PCR amplification and both fragments and vectors (pBT3-N and pPR3-N) were digested by SfiI. Other plasmids were all constructed using NEBuilder HIFI DNA Assembly Master Mix (New England Biolabs). The fragments were also prepared by PCR amplification and vectors were linearized by restriction endonuclease double digestion.

Full length GS3.1 genome sequence from the translation start codon to the last codon before the stop codon from HJX were cloned into pCAMBIA1300 under the control of rice ubiquitin promoter to create the GS3.1 overexpression construct. The sequence from 2.5 kbp upstream of the translation start codon of GS3.1 to 1.5 kbp downstream of the stop codon of GS3.1 from HJX was cloned into

pCAMBIA1300 for the genomic complementation construct. The GUS plus gene promoted by GS3.1 promoter region (from 2.5 kbp upstream to the last base-pair before the translation start codon of GS3.1) of HJX allele was cloned to pCAMBIA1300 for the ProGS3.1<sup>HJX</sup>:GUS construct.

All plasmids used in this study were confirmed by sequencing. PCR primers for fragment amplification were listed in Supplementary Data 2 together with restriction endonuclease digestion sites.

**Plant transformation.** Agrobacterium tumefaciens-mediated transformation of rice calli was performed using strain EHA105[86]. Constructs were transferred into EHA105 Agrobacterium cells, which were grown in Yeast Mannitol Broth (YEB) medium at 30 °C overnight and then co-cultivated with rice calli for three days. The transgenic plants were then screened on 25 μg·mL⁻¹ hygromycin. All transgenic plants were identified by PCR amplification of the hygromycin resistant gene and GS3.1 CRISPR/Cas9 transgenic plants were additionally identified by sequencing of the GS3.1 target region.

**Histochemical analysis and electron microscopy**[24,87,88]. Spikelet hulls before heading were fixed in FAA (50% ethanol, 5% glacial acetic acid, and 5% formaldehyde) and held *in vacuo* for 15 min for three to four times, then dehydrated using an ethanol series (30%, 50%, 70%, 80%, 90%, 95%, 100%, and 100% ethanol for 15 min each). Tissues were transferred to acetone and then embedded in Eponate 12 Resin (TED PELLA) and sliced into 8-μm or 6-μm thin sections with a rotary microtome (Leica). Sections of spikelet hulls were stained by Toluidine Blue (TBO), and the prepared tissue sections were observed under a light microscope (Carl Zeiss). For electron microscopy scanning, the spikelet hulls before heading were dried in a critical point drier (Leica) after dehydration, and mature grains were dried at 42 °C for 3–5 d after harvesting. Then the spikelet hulls were gold sputter coated, and imaged under a scanning electron microscope (JEOL and Carl Zeiss). Image J software was used to measure cell size and cell number[89,90].

**GUS staining.** GUS staining was performed according to the previous method[91]. Tissues were soaked in acetone at 4 °C for 20 min and washed in PBS (50 mM, pH 7.0), then transferred to GUS staining buffer (PBS, 0.5 mM K-Fe-Nc, 0.1% Triton X-100, 20% methanol, and 0.6 mg·mL⁻¹ X-Gluc) and held in vacuo for 15 min for twice. After that, samples were stained overnight in GUS staining buffer at 37 °C, and cleared in 95% ethanol to remove chlorophyll. The prepared tissues were then observed under a stereomicroscope and Leica Application Suite (Version V4.2) was used for image collection. For observing sections, cleared tissues were then sectioned and imaged as indicated above.

**Phylogenetic relationship analysis.** Multiple sequence alignments were performed in MEGA6 by MUSCLE under default parameters[92]. Phylogenetic analysis was carried out using the maximum-likelihood method in MEGA. LG with frequencies and gamma distributed (G) model and a complement deletion treatment for missing data (gaps) with a 1000 replications bootstrap analysis were used.

**Prediction of GS3.1 three-dimensional (3D) structure and molecular docking.** Three-dimensional (3D) structural models of GS3.1 were produced by Phyre2 mainly according to the previously described AtDTX14 structure (c5y50A) using default parameters[43,93]. The prediction of ligand-binding sites for the interaction with target molecules was made by molecular docking using the GEMDOCK software (BioXGEM) under the default parameters for stable docking[94].

**E. coli mutant growth complementation assay.** The growth complementation assay using the drug-sensitive *E. coli* strain (BW25113Δacrb) was performed as described previously[41,42]. The full-length coding sequence of GS3.1 was cloned into pMAL-c5X (pMAL, New England Biolabs). The constructed pMAL-GS3.1 and pMAL were used to transform the Δacrb strain. Transformants and the wild-type BW25113 strain were streaked on LB plates containing 0.25 mM isopropyl β -D-thiogalactoside (IPTG), and 0.035 μg·mL⁻¹ norfloxacin (Sigma) was added to the plate for drug resistance testing.

**Widely-targeted metabolomics assay.** Widely-targeted metabolomics was performed by Metware (Wuhan, China) as previously described[6]. The freeze-dried plant samples were ground using a mixer mill (MM 400, Retsch) with zirconia beads at 30 Hz for 1.5 min. The powder was weighed (100 mg) and extracted with 1 mL 70% aqueous methanol overnight at 4 °C. After centrifugation at $10,000 \times g$ for 10 min, the sample extracts were absorbed (CNWBOND Carbon-GCB SPE Cartridge, 250 mg, 3 ml; ANPEL, Shanghai, China) and filtered (SCAA-104, 0.22 μm pore size; ANPEL). Then the extracts were analyzed using an LC-ESI-MS/MS system.

**Lignin content measurement.** For measuring lignin content, the cell walls were first isolated and then lignin was extracted from the isolated cell walls as previously described[95,96]. In brief, tissue powder was extracted with 70% ethanol at 70 °C for 1 h 3 times, the supernatant was discarded, and the residue was dried at 50 °C to a

consistent weight. The alcohol-insoluble residue (largely cell wall material) was weighed (10–15 mg) and hydrolyzed with 0.3 mL thioglycolic acid and 1.5 mL 2 M HCl at 95 °C for 4 h. Centrifugations at 15,000 × g for 15 min were performed for precipitate collection and the material was washed three times with distilled water. Then the precipitate was extracted with 0.5 M NaOH at 20 °C with shaking for 16 h twice (1 mL and 400 μL NaOH). The supernatants were combined after centrifugation at 15,000 × g for 15 min each time. The combined supernatant was acidified with 400 μL concentrate HCl at 4 °C for 4 h, collected by centrifugation at 15,000 × g for 20 min and dissolved in 0.5 M NaOH. The absorbance of samples was measured at 280 nm ($OD_{280}$) against a 0.5 M NaOH blank using a spectrometer (Beckman Coulter). The relative content of lignin was determined as $OD_{280} \cdot g^{-1}$ of the weighed cell wall powder.

**Split-ubiquitin assay.** The split-ubiquitin assay was performed using the DUALmembrane pairwise interaction kit (Dualsystems Biotech)[16,66]. The membrane protein GS3.1 was cloned into the *pBT3-N* bait vector and *OsATG8b* was cloned into the *pPR3-N* prey vector. The yeast reporter strain NMY51 was used and *pOst1-NubI* was used as the positive control prey vector.

**Luciferase complementation assay.** The luciferase complementation assay was conducted as described previously[97]. *GS3.1* was fused to the N-terminus of luciferase (nLUC) and *OsATG8b* was fused with C-terminus of luciferase (cLUC). Agrobacterium tumefaciens-mediated transient transformation was performed by injection into leaves of *N. tabacum* using strain GV3101. Luciferin (Promega) and UVP ChemStudio PLUS (Analytik Jena) were used for fluorescence excitation and imaging.

**Open resource data acquisition.** Rice genomic information was acquired from RGAP (http://rice.plantbiology.msu.edu/). *Arabidopsis* genomic information was retrieved from TAIR (https://www.arabidopsis.org/). Compound structures were downloaded from PubChem (https://pubchem.ncbi.nlm.nih.gov/).

**Statistics and reproducibility.** The difference between two groups were assessed by Student's *t*-tests. The values represented in mean ± s.d. Statistically significant differences among multiple groups were evaluated by ANOVA followed by a Tukey's test. Details of each statistical test are indicated in the figure legends.

**Reporting summary.** Further information on research design is available in the Nature Research Reporting Summary linked to this article.

## Data availability

Source data of figures and Supplementary figures are provided in Supplementary Data 1. The primer information is provided in Supplementary Data 2. The data that support the findings of this study are available from the corresponding author upon reasonable request.

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

## Acknowledgements

We thank Professor Guiquan Zhang (South China Agricultural University) for providing CSSL plant materials. We thank Min Shi (CAS Center for Excellence in Molecular Plant Sciences, Shanghai Institute of Plant Physiology and Ecology, CAS) for technical supports of transgenic assay. We thank Jiqin Li and Zhiping Zhang (CAS Center for Excellence in Molecular Plant Sciences, Shanghai Institute of Plant Physiology and Ecology, CAS) for technical supports of histochemical analysis and electron microscopy scanning. We thank Professor Yaoguang Liu (South China Agriculture University) for donation of CRISPR/Cas9 plasmids. We thank Professor Fei Yu (Northwest A & F University) for donation of BW25113Δacrb and BW25113 E. coli strain. This work was supported by the grants from Laboratory of Lingnan Modem Agriculture Project (NT2021002), Chinese Academy of Sciences (XDB27010104, QYZDY-SSW-SMC023, 159231KYSB20200008), National Natural Science Foundation of China (31788103), the Shanghai Science and Technology Development (18JC1415000), CAS-Croucher Funding Scheme for Joint Laboratories and National Key Laboratory of Plant Molecular Genetics.

## Author contributions

H.X.L. conceived and supervised the project, and H.X.L. and Y.M.Z. designed the experiments. Y.M.Z. performed most of the experiments. H.X.Y., W.W.Y., J.X.S., N.Q.D., T.G., Y.K., Y.H.X., H.Z., Y.B.Y., Y.C.L., H.Y.Z., Z.Q.L., S.Q.G., J.J.L., B.L., X.R.M., Y.J.C., J.J.Y., and H.X.L. performed some of the experiments. Y.M.Z. and H.X.L. analyzed data and wrote the manuscript.

## Competing interests

The authors declare no competing interests.
