## [Transparent Peer Review File · Communications Biology]

Reviewers' comments:

Reviewer #1 (Remarks to the Author):

Author and team did a good research work in field, lab and as a current manuscript but there are many suggestions by the reviewer in attached file. There were many grammatical errors in manuscript as author and team belong to non-English background so, It will be appreciate to revise the manuscript with a English expert.

Hope an improved version of manuscript.

Reviewer #2 (Remarks to the Author):

The present study entitled "A rice QTL GS3.1 regulates grain yield through metabolic-flux distribution between flavonoid and lignin metabolons without affecting stress tolerance" provides a detailed and comprehensive analysis of GS3.1 QTL, which was associated with grain size development in rice. GS3.1 which encodes multidrug and toxic compounds extrusion transporter protein regulates grain size by directing the transport of p-coumaric acid from the 28 p-coumaric acid biosynthetic metabolon to the flavonoid biosynthetic metabolon. overall the manuscript is well presented and analyzed in details. statistical analysis is valid and consistence with the observed results. I think the manuscript need a little bit polishing in term of English usage. In addition, authors mentioned that "a set of chromosome segment substitution lines (CSSLs) were constructed with an African..." can authors provides cytogenetic image for such chromosol substitution?

Reviewer #3 (Remarks to the Author):

Comments to author

1. Authors identified a gene underlying the QTL GS3.1 and identified function of this gene for enhancing the grain size. They proved that larger grain size caused by this QTL enhanced grain yield without affecting tillers number and height. Therefore authors reported that this QTL regulate grain yield in rice. However, authors have conducted this study in one background, while effect of this QTL may be changed in other background and enhanced seed size caused by this QTL may be negatively affect other yield contributing traits. Therefore title of manuscript may be changed by focusing on grain size rather than yield.
2. Authors do not studied the NILs having the targeted QTL under stress conditions. Why authors mentioned "without affecting stress tolerance" in the title of MS?
3. Line 147-155, NILs GS3.1HP has more number tillers along with seeds size while though gene editing authors mentioned that identified gene that regulate seed size within GS3.1 locus has no effect PH and tiller number. Are these results not contradictory? Explain!
4. Line number 179 "grain weight before 15 days after flowering". How we can measure grain weight before flowering?
5. Arabidopsis is a scientific name. Why it has be written in normal font in throughout the manuscript.
6. In Figures, line names either should be in small cap or large cap.
7. Line number 184, "Spikelet nulls" or "Spikelet hulls".
8. There are several typographical mistakes that are needed to take care.

Point-by-point Response to Reviewers

Dear Reviewers,

We are very grateful for the three referees' comments on our manuscript. You provided valuable insights, critical comments and thoughtful suggestions, all of which were very helpful for revising and improving our manuscript. Based on your comments and suggestions, we performed further experiments (two Supplementary Figures and two Supplementary Tables were added to the new manuscript) and made careful modifications to the original manuscript. We hope that the revised manuscript is more satisfactory. The main changes are highlighted in yellow in the revised version of the manuscript. Detailed descriptions of the revisions and responses to the reviewers' comments are provided below.

Reviewer #1 (Remarks to the Author):

Author and team did a good research work in field, lab and as a current manuscript but there are many suggestions by the reviewer in attached file. There were many grammatical errors in manuscript as author and team belong to non-English background so, It will be appreciate to revise the manuscript with a English expert.

Hope an improved version of manuscript.

Response:

Thank you for all the helpful and valuable comments in the attached file. We tried our best to improve our manuscript according to these comments. We picked out some comments which we consider important for further illustration.

1. Line 89 of the attached file

Please shift these lines to first paragraph of material and method under subheading "Plant materials and growth conditions"

Not here in result section

Response:

We think that the parent information is necessary for readers to understand our work. The readers may be better to get this information in the result part. So, we reserved the parent information here but did some simplification (Line 81-86 in new manuscript).

2. Line 148 of the attached file

please add the table number of agronomic data

Response:

The data has been shown in the histograms (**Fig. 2b-i**). Also, we now add these data in new **Supplementary Table 3**.

3. Line 185 of the attached file

Please quantify the values in brackets

Response:

The data has been shown in the histograms (**Fig. 2l, m, o, p**). Also, we now add these data in new **Supplementary Table 4**.

4. Line 440, 446 and 452 of the attached file

Response:

Here, we consider that the *p*-coumaric acid, flavonoid and lignan biosynthesis metabolons are formed by LLPS (liquid-liquid phase separation). Because there's some evidence that membrane-less metabolic enzymes may form in this way^{1,2}. Then we consider the relationship between GS3.1 and the metabolons. We find that one kind of autophagy-related protein (ATG8) have reported taking part in the formation of biomolecular condensates formed by LLPS³. Meanwhile, this ATG8 protein interacted with ABS4 (the ortholog of GS3.1 in *Arabidopsis*)⁴. Moreover, the subcellular localization of ABS4 was the same as the *p*-coumaric acid, flavonoid and lignan biosynthesis metabolons^{4,5}. So that, we speculate that GS3.1 may work as a transporter between metabolons with the assist of ATG8.

1. Xie, D. *et al.* Phase separation of SERRATE drives dicing body assembly and promotes miRNA processing in Arabidopsis. *Nat. Cell Biol.* **23**, 32–39 (2021).
2. Wunder, T. & Mueller-Cajar, O. Biomolecular condensates in photosynthesis and metabolism. *Curr. Opin. Plant Biol.* **58**, 1–7 (2020).
3. Wilfling, F. et al. A Selective Autophagy Pathway for Phase-Separated Endocytic Protein Deposits. *Mol. Cell* 80, 764-778.e7 (2020).
4. Jia, M. et al. Noncanonical ATG8–ABS3 interaction controls senescence in plants. *Nat. Plants* 5, 212–224 (2019).
5. Achnine, L., Blancaflor, E. B., Rasmussen, S. & Dixon, R. A. Colocalization of L-phenylalanine ammonia-lyase and cinnamate 4-hydroxylase for metabolic channeling in phenylpropanoid biosynthesis. *Plant Cell* 16, 3098–3109 (2004).

5. Line 481 of the attached file

Please add a subheading 'Conclusion'

Add these lines there with following points

1. What were main key outcome of the current study (in 2 or 3 lines only)
2. What was take home message for readers from the current study
3. What was way forward for the future research in continuation of current research work.

Response:

We consider that the format of *Communication biology* does not have a conclusion part. So, according to your comments, we write these points as a new paragraph at the end of the discussion (Line 466-474 in new manuscript).

6. Line 494 of the attached file

Author and team, please revise make clear in manuscript about " GS3.1."

If this QTL (GS3.1.) was identified in previous study, then make clear in introduction section

Otherwise If this QTL (GS3.1.) was identified in the current study then please use this term (GS3.1.) in Result and discussion section only. Please use term “Grain size” in introduction and Material and method.

Using “GS3.1.” in introduction and Material and method, will make confusion to readers.

Response:

The QTL (*GS3.1*) was identified in the current study, so according your comments, we avoided to refer to “*GS3.1*” in the introduction part. But, in the method part, it is necessary to mention “*GS3.1*” when talking about the certain experiments like the plasmid construction. So, we kept the terms (*GS3.1*) there and added a definition at line 490-492 in our new manuscript. In this way, readers will be easy to understand the methods.

7. Line 506 of the attached file

Please add the experiment was conducted in open field or pot experiment. Or in controlled condition.

If it was open field then add min-max temperature, humidity, and rain fall

If it was in controlled condition then also temperature, RH and Light intensity

What was soil type, pH

What was agronomic practices applied (use a reference)

Fertilizer and pesticide doses

What was seed sowing date/ year?

(if transplanting happened then what was nursery date and after how many days nursery was transplanted?)

What was crop duration / harvesting date ?

If any experimental design was applied then which one, how many replications and how many plants per replication?

Response:

We are sorry that we cannot get the data of humidity in the certain months we grew our plant materials. However, we tried to get the geographic coordinate, altitude, soil type and soil pH data of the experimental field, and the temperature and rainfall during our growing period (Line 496-504 in new manuscript). We think these data can almost reveal the growth condition of the plant materials. The information of replication has been shown in per figure legend. Thank you for your comments.

8. Line 530 of the attached file

Please make clear 3 or 4????

Response:

We changed the sentence to “Actin (*LOC_Os03g50885*) was used for normalization and either three or four biological replicates were used for one analysis” (Line 529-531 in new manuscript). We mean that a part of qRT-PCR experiment used 3 biological replications, and others used 4 biological replications. We give clear indication of

biological replication numbers in corresponding figure legends.

9. Line 530 of the attached file

Please add the reference of this protocol

Response:

The company (Dualsystems Biotech) do not list their publications. But some previous studies used this system^{1,2}.

1. Tian, W. et al. A molecular pathway for CO₂ response In Arabidopsis guard cells. Nat. Commun. 6, 6057 (2015).
2. Duan, P. et al. Regulation of OsGRF4 by OsmiR396 controls grain size and yield in rice. Nat. Plants 1, 15203 (2015).

Thank you again for all your good comments.

Reviewer #2 (Remarks to the Author):

The present study entitled "A rice QTL GS3.1 regulates grain yield through metabolic-flux distribution between flavonoid and lignin metabolons without affecting stress tolerance" provides a detailed and comprehensive analysis of GS3.1 QTL, which was associated with grain size development in rice. GS3.1 which encodes multidrug and toxic compounds extrusion transporter protein regulates grain size by directing the transport of p-coumaric acid from the

28 p-coumaric acid biosynthetic metabolon to the flavonoid biosynthetic metabolon. overall the manuscript is well presented and analyzed in details.

statistical analysis is valid and consistence with the observed results.

I think the manuscript need a little bit polishing in term of English usage.

In addition, authors mentioned that "a set of chromosome segment substitution lines (CSSLs) were constructed with an African..." can authors provides cytogenetic image for such chromosol substitution?

Response:

Thank you for your comments. We revise the language of this manuscript and hope the revised manuscript can be more satisfactory. We consider that the cytogenetic image for all chromosol substitution lines is quite large and not very closely related to our current work. So that, we instead show the cytogenetic image for the certain line we use for mapping *GS3.1*, HPC078, and two near isogenic lines of *GS3.1* (new **Supplementary Figure 1**). The fragment from HP in chromosome 6 is reserved owing to this fragment containing a hybrid sterility locus. We make sure that the genetic background in NIL-*GS3.1*^{HJX} and NIL-*GS3.1*^{HP} were the same. Thank you again for your good comments.

Supplementary Figure 1. Cytogenetic image of genetic materials of *GS3.1*. The cytogenetic image of the CSSL containing *GS3.1* (a), NIL-*GS3.1*^{HJX} (b) and NIL-*GS3.1*^{HP} (c).

Reviewer #3 (Remarks to the Author):

Comments to author

1. Authors identified a gene underlying the QTL *GS3.1* and identified function of this gene for enhancing the grain size. They proved that larger grain size caused by this QTL enhanced grain yield without affecting tillers number and height. Therefore authors reported that this QTL regulate grain yield in rice. However, authors have conducted this study in one background, while effect of this QTL may be changed in other background and enhanced seed size caused by this QTL may be negatively affect other yield contributing traits. Therefore title of manuscript may be changed by focusing on grain size rather than yield.

Response:

Thank you for this suggestion. We change the title to "**A rice QTL *GS3.1* regulates grain size through metabolic-flux distribution between flavonoid and lignin metabolons without affecting stress tolerance**".

2. Authors do not studied the NILs having the targeted QTL under stress conditions. Why authors mentioned "without affecting stress tolerance" in the title of MS?

Response:

Thank you for your insightful comment. We draw this conclusion mainly based on the metabonomic data. The phenylpropane metabolism in leaves were no significant difference between two near isogenic lines of *GS3.1* (new **Supplementary Figure 11** which was **Supplementary Figure 10** in old manuscript). To confirm this, we treated these two NILs with 100 mM NaCl for 25 days and recover for 6 days, and the results indicated that there is no difference between NIL-*GS3.1*^{HP} and NIL-*GS3.1*^{HJX} in salt tolerance (the photograph of recovered seedlings and the statistical data of the survival ratio are shown in new **Supplementary Figure 12**).

Supplementary Figure 12. There is no difference between NIL-GS3.1^{HP} and NIL-GS3.1^{HJX} in salt tolerance. a, The photograph of the growth status of NIL-GS3.1^{HP} and NIL-GS3.1^{HJX} seedlings after a 25 days' NaCl treatment and 6 days' recovery. **b,** The survival ratio of the growth status of NIL-GS3.1^{HP} and NIL-GS3.1^{HJX} seedlings after a 25 days' NaCl treatment and 6 days' recovery. The source data underlying Supplementary Figure 12b are provided as Source Data file.

3. Line 147-155, NILs GS3.1HP has more number tillers along with seeds size while though gene editing authors mentioned that identified gene that regulate seed size within GS3.1 locus has no effect PH and tiller number. Are these results not contradictory? Explain!

Response:

Thank you for your insightful comment. We also notice these inconsistent results and confirm that this difference on tiller number was real existence. So, we think this may because of the different mutant type of *GS3.1* (the natural mutant in NIL-GS3.1^{HP} and complete knock-out in KO-GS3.1).

4. Line number 179 "grain weight before 15 days after flowering". How we can measure grain weight before flowering?

Response:

Thank you for your comment. We change the sentence to "**However, there was no**

significant difference in grain weight from the flowering day to 15 days after flowering" (Line 170-171 in new manuscript). We mean that the grain weight showed no significant difference in three timings from the flowering day to 15 days after flowering (5 days after flowering, 10 days after flowering and 15days after flowering).

5. Arabidopsis is a scientific name. Why it has be written in normal font in throughout the manuscript.

Response:

Thank you for your comment. We changed them in italic in the new manuscript.

6. In Figures, line names either should be in small cap or large cap.

Response:

Thank you for your comment. We changed line names in the small cap to large cap (**Fig. 1**, new **Supplementary Figure 2** and new **Supplementary Figure 4**).

7. Line number 184, "Spikelet nulls" or "Spikelet hulls".

Response:

Thank you for your comment. We changed "Spikelet nulls" to "Spikelet hulls" in the new manuscript (Line 175 in new manuscript).

8. There are several typographical mistakes that are needed to take care.

Response:

Thank you for your comments. We revise the language of this manuscript and hope the revised manuscript can be more satisfactory.

Thank you again for your constructive comments.

Thanks a lot for all Reviewers' good comments. It's quite helpful for us to improve our manuscript.

REVIEWERS' COMMENTS:

Reviewer #3 (Remarks to the Author):

Authors have satisfactorily addressed the comments.